# Overlapping Systemic Proteins in COVID-19 and Lung Fibrosis Associated with Tissue Remodeling and Inflammation

**DOI:** 10.3390/biomedicines12122893

**Published:** 2024-12-19

**Authors:** Barbora Svobodová, Anna Löfdahl, Annika Nybom, Jenny Wigén, Gabriel Hirdman, Franziska Olm, Hans Brunnström, Sandra Lindstedt, Gunilla Westergren-Thorsson, Linda Elowsson

**Affiliations:** 1Lung Biology Unit, Department of Experimental Medical Science, Lund University, 221 84 Lund, Sweden; barbora.svobodova@med.lu.se (B.S.); jenny.wigen@med.lu.se (J.W.); gunilla.westergren-thorsson@med.lu.se (G.W.-T.); 2Department of Cardiothoracic Surgery and Transplantation, Skåne University Hospital, 222 42 Lund, Sweden; gabriel.hirdman@med.lu.se (G.H.); franziska.olm@med.lu.se (F.O.); sandra.lindstedt@med.lu.se (S.L.); 3Wallenberg Center for Molecular Medicine, Lund University, 221 84 Lund, Sweden; 4Department of Clinical Sciences, Lund University, 221 84 Lund, Sweden; hans.brunnstrom@med.lu.se

**Keywords:** COVID-19, fibrosis, IPF, biomarkers, DCN, TNFRSF12A, MCP-3, HGF

## Abstract

**Background/Objectives:** A novel patient group with chronic pulmonary fibrosis is emerging post COVID-19. To identify patients at risk of developing post-COVID-19 lung fibrosis, we here aimed to identify systemic proteins that overlap with fibrotic markers identified in patients with idiopathic pulmonary fibrosis (IPF) and may predict COVID-19-induced lung fibrosis. **Methods:** Ninety-two proteins were measured in plasma samples from hospitalized patients with moderate and severe COVID-19 in Sweden, before the introduction of the vaccination program, as well as from healthy individuals. These measurements were conducted using proximity extension assay (PEA) technology with a panel including inflammatory and remodeling proteins. Histopathological alterations were evaluated in explanted lung tissue. **Results:** Connecting to IPF pathology, several proteins including decorin (DCN), tumor necrosis factor receptor superfamily member 12A (TNFRSF12A) and chemokine (C-X-C motif) ligand 13 (CXCL13) were elevated in COVID-19 patients compared to healthy subjects. Moreover, we found incrementing expression of monocyte chemotactic protein-3 (MCP-3) and hepatocyte growth factor (HGF) when comparing moderate to severe COVID-19. **Conclusions:** Both extracellular matrix- and inflammation-associated proteins were identified as overlapping with pulmonary fibrosis, where we found DCN, TNFRSF12A, CXCL13, CXCL9, MCP-3 and HGF to be of particular interest to follow up on for the prediction of disease severity.

## 1. Introduction

The COVID-19 pandemic has caused severe aftermaths, with pulmonary fibrosis being considered an emerging problem. Around 20% of COVID-19 patients have developed severe symptoms, usually appearing after a week of an initial sick period, where about 5% of these patients have presented with critical conditions such as acute respiratory distress syndrome (ARDS). These conditions result from the massive immune response involving a cytokine storm [1], leading to an imbalance of the renin–angiotensin system (RAS) and the dysregulation of various cytokines [2,3]. RAS is primarily known for its role in regulating blood pressure and fluid balance, implicated in COVID-19 patients with common complications of coagulopathy [4] and endotheliopathy [5]. In addition, the cytokine storm contains an excess of pro-inflammatory markers including interleukins, monocyte attractant factors and tumor necrosis factor (TNF) [6,7], several factors that have been described in chronic interstitial lung diseases such as idiopathic pulmonary fibrosis (IPF). IPF is thought to be the result of repeated microinjuries of the epithelium, leading to a dysregulated epithelial–mesenchymal crosstalk that in turn results in an aberrant wound healing response [8,9].

We and others have demonstrated in vitro how a distorted microenvironment promotes the progression of disease [10]. Severe cases of COVID-19 typically exhibit diffuse alveolar damage (DAD) characterized by fibrin exudates, hyaline membrane formation, hyperplasia and a loss of alveolar type II cells, the disruption of the basement membrane and thickened alveolar walls. These changes can result in alveolar collapse, edema, inflammation and fatal dyspnea [11,12]. In IPF, fibroblasts are thought to be the main effector cell contributing to the accumulation of excessive extracellular matrices (ECMs) in the lungs, activated by pathways including TNF and transforming growth factor beta (TGF-β) and by epithelial-to-mesenchymal transition (EMT). Additional studies have implicated monocyte chemotactic protein-3 (MCP-3) and Chemokine (C-X-C motif) ligand 13 (CXCL13) as biomarkers for the pathogenesis of IPF, suggesting their roles in fibroblast recruitment and activation [7,13,14]. ECM-associated proteins such as decorin (DCN) and periostin (POSTN) are both involved in tissue remodeling, DCN in fibrillogenesis and POSTN in fibroblast activation and collagen deposition. The long-term risk of developing pulmonary fibrosis increases with the severity of initial diseases including ARDS and has been seen to be more pronounced in patients who have been subjected to mechanical ventilation [15]. Ravaglia et al. recently reported on histological patterns in patients with persistent symptoms after COVID-19 infection. Classified into three clusters, two were related to chronic fibrosing and different grades of lung injury for usual interstitial pneumonia (UIP) and DAD [16].

Identifying patients at risk of developing chronic pulmonary fibrosis is important for managing and intervening with the underlying pathological pathways [3,7,9,12,17]. Smoking, identified as a risk factor in the onset of lung fibrosis [18] has also been noted as a contributing factor to all-cause mortality in COVID-19 infections [19]. We hypothesized that early alterations in systemic cytokines and ECM components associated with remodeling may be indicative of the risk of developing chronic manifestations of pulmonary fibrosis. In this study we identified potential biomarkers in hospitalized COVID-19 patients, included prior to the vaccination program, with both moderate and severe symptoms, that overlap with previous findings linked to pulmonary fibrosis. The biomarkers were found to be associated with both inflammation and tissue remodeling. Among the identified proteins, we found elevated systemic levels of remodeling proteins decorin (DCN), tumor necrosis factor receptor superfamily member 12A (TNFRSF12A) and hepatocyte growth factor (HGF), and chemoattractant factors MCP-3 and CXCL13 in pre-vaccine plasma samples from COVID-19 patients, which supports previous findings of these molecules in patients with IPF and pulmonary fibrosis post COVID-19 [20,21,22,23,24].

## 2. Materials and Methods

### 2.1. Study Design

Plasma samples collected from hospitalized pre-vaccine COVID-19 patients were analyzed and divided into two groups: severe and moderate. The two groups were compared to healthy subjects that were age- and gender-matched. Selected proteins of interest were further analyzed by histology and immunohistochemistry in explanted lung tissue from COVID-19 patients and compared to IPF patients and healthy subjects. Appendix A presents patient and healthy control characteristics.

### 2.2. Plasma from COVID-19 Patients and Controls

Plasma samples were collected (in EDTA tubes) two weeks after hospital admission during 2020 prior to the initiation of COVID-19 vaccination programs. Patients with severe COVID-19 and ARDS were treated via mechanical ventilation (n = 8), and patients with moderate symptoms (n = 8) were treated with supplemental oxygen. Patients treated via mechanical ventilation had a PaO_2_/FiO_2_ ratio between 80 and 200 at the time of blood sample collection, equal to severe-to-moderate ARDS using the Berlin definition of ARDS [25]. None of the patients on mechanical ventilation had any CO_2_ retention. Patients treated with oxygen support other than mechanical ventilation had a P/F ratio above 200 but below 300 at the time of blood sample collection, equal to mild ARDS using the Berlin definition of ARDS. None of the patients had any CO_2_ retention. As the control group, plasma samples from age- and gender-matched healthy subjects were used (n = 7). The plasma samples were analyzed in an immuno-oncology panel (encompassing inflammatory and remodeling proteins) of 92 proteins using proximity extension assay (PEA) technology (Olink AB, Uppsala, Sweden). PEA is a highly sensitive method that uses antibody pairs with DNA tags to detect and quantify multiple proteins simultaneously via qPCR. The panel used in this study was chosen to match the one employed in our previous biomarker research on IPF.

### 2.3. Explanted Human Lung Tissue

Explanted distal lung tissue specimens derived from post-COVID-19 patients, IPF patients and lung donors were used for histological evaluation. Specimens from explanted lungs were collected from three post-COVID-19 patients (male n = 1, age 61; female n = 2, age = 63, and 64; all ex-smokers). One of the post-COVID-19 patients was treated with extracorporeal membrane oxygenation for 5 months before the lung transplantation. The other two patients developed interstitial lung disease, out of which one of them had symptoms of COPD. The two IPF patients included in this study (female n = 1, age 65; male n = 1, age 57; both ex-smokers) were diagnosed based on ATS and ERS criteria [26]. Three healthy lungs from organ donors were included (males n = 3, age 62, 66 and 68) out of which two were ex-smokers and one had never smoked.

### 2.4. Histology and Immunohistochemistry

Formalin-fixed paraffin-embedded distal lung tissue with 1–3 tissue blocks per individual were antibody-labeled with HRP-DAB immunohistochemistry for periostin (POSTN) and decorin (DCN), with corresponding tissue sections stained with hematoxylin and eosin (HE) for the visualization of tissue morphology and pathological features. In short, following deparaffinization, sections were treated with heat-induced antigen retrieval at a low pH, using Flex target retrieval solution (cat no. K8005, Agilent Dako, Santa Clara, CA, USA). The sections were stained with primary antibodies for DCN (1:1000, cat no. HPA003315, Atlas antibodies, Bromma, Sweden) and POSTN (1:1000, ab79946, Abcam, Cambridge, UK) together with Envision dual link system-HRP (DAB) (cat no. K4065, Dako) and counterstained with hematoxylin before dehydration and mounting with Pertex. The sections were scanned with the VS120 virtual microscopy slide scanning system (VS120-L100-FL080, Olympus, Tokyo, Japan). Representative images were acquired using the OlyVIA software 3.3 (Olympus) Qupath 0.3.0 (The University of Edinburgh, UK) [27] and ImageJ 1.53j (National Institutes of Health, Bethesda, MD, USA). The morphology of tissue sections was assessed by using the modified Russell–Movat pentachrome stain kit (cat. No. KSC-L53PIA-1, Nordic Biosite, Täby, Sweden) according to the manufacturer’s protocol.

### 2.5. Statistical Analysis

The data from the PEA of the plasma samples were presented as normalized protein expression (NPX) values for each protein on a log2 scale, i.e., a difference of 1 NPX between proteins equals a doubling of protein concentration. In the analyzed samples, six proteins with a detectability lower than 67% (>33% of limit of detection per protein) were excluded (IL13, IL-1α, IL-2, IL33, CD28, IL4), resulting in the inclusion of 86 proteins (93.5%) to be evaluated. One-way ANOVA with Tukey’s multiple comparisons test with a cut-off at a mean group difference >1 NPX was used to analyze statistical differences between groups. We categorized the proteins, as described in Kalafatis et al. (2021) [28], according to three main biological functions related to inflammation and chemotaxis (I), tissue remodeling (R) or proteins with overlapping functions (O).

## 3. Results

### 3.1. Altered Proteins in Plasma Samples from COVID-19 Patients and Healthy Controls

The majority of included COVID-19 patients were men in both the severe and moderate group, of which two died during their hospital visit. The healthy controls were age- and gender- matched (Table 1); however, smoking status for most of the patients with severe COVID-19 was not available.

Comparing plasma samples from patients with severe COVID-19 with those from healthy controls revealed 42 significantly altered proteins with a >1 NPX mean difference. Out of these, 23 proteins were also significantly elevated in patients with moderate COVID-19 in comparison to healthy controls. Fifteen proteins were significantly increased in the severe group in comparison to healthy controls, of which pleiotrophin (PTN), adhesion G protein-coupled receptor G1 (ADGRG1) and C-C motif chemokine ligand 19 (CCL19) were the top three proteins most elevated in patients with severe COVID-19 (Table 2). As seen in our previous studies on lung fibrosis (5, 8), DCN was elevated in severe COVID-19 patients (Table 2, Figure 1).

In patients with moderate COVID-19, but not with severe illness, four proteins involved in inflammatory and chemotactic processes were significantly reduced in comparison to healthy controls (Table 2). Distinguishing patients with severe and moderate COVID-19, seven proteins were significantly altered (>1 NPX mean difference) (Table 2) including CXCL13 and C-X-C motif chemokine ligand 9 (CXCL9), where TNF receptor superfamily member 12A (TNFRSF12A) was the most increased (Figure 1), highlighting the involvement of the TNF-a pathway. MCP-3 and HGF were significantly increased (>1 NPX mean difference) in all patient groups, incrementing from healthy to moderate to severe COVID-19 (Figure 1). All significantly altered proteins with a >1 NPX mean difference between COVID-19 groups and healthy controls are shown in Appendix A. It is noteworthy that two patients with severe COVID-19 who died during their hospital visit exhibited additional elevated plasma levels of CCL19, placenta growth factor (PGF) and carbonic anhydrase IX (CAIX), proteins mainly involved in tissue remodeling processes (Supplement Appendix A).

### 3.2. Tissue Morphology and Protein Expression in Distal Lung in Severe Post-COVID-19 and IPF

Tissue sections from post-COVID-19 and IPF patients demonstrated an accumulation of dense connective tissue rich in collagens, proteoglycans and disrupted parenchymal structures with inflammatory cell infiltrates (Supplement Appendix A). The airways and alveolar structures were filled with mucus and inflammatory cells (in particular macrophages). Furthermore, fibroblast foci formation in IPF and the accumulation of fibroblasts and myofibroblasts in post-COVID-19 resembling the IPF fibroblast foci as well as possible alveolar edema and alveolar type II hyperplasia were observed. With computed tomography, one of the patients with COVID-19 showed consolidated parenchymal changes in their lungs with severe ARDS and fibrotic development (Supplement Appendix A). Decorin, known for its role in collagen fibrillogenesis, showed increased expression with clear localization to the subepithelial regions of bronchioles (Figure 2), consistent with previous findings in IPF lung tissue [10]. Similarly, periostin, which is involved in fibroblast recruitment at active fibrotic sites, also exhibited elevated expression in the subepithelial regions of bronchioles. Moreover, periostin was localized to fibroblastic foci in IPF lungs and in similar structures within post-COVID-19 lung tissue (Figure 2G–J). A shared spatial expression pattern of DCN was seen in both post-COVID-19 and IPF with elevated staining intensity, corresponding to the increased deposition of extracellular matrices. DCN was detected in the vascular adventitia (Figure 2K–L), and both DCN and POSTN were highly expressed in the visceral pleura, where DCN was also expressed in mesothelial cells (Figure 2N–P, Appendix A). In heavily remodeled areas with dense connective tissue and cell infiltration in post-COVID-19, DCN was seen to be embedded in the ECM with fiber-like formation, while in IPF, DCN appeared more fragmented. DCN and POSTN were also found in regions of honeycomb cysts, a typical pathological pulmonary feature in IPF (Appendix A).

## 4. Discussion

The immune response of COVID-19-affected lungs results not only in inflammatory activity but also in long-term structural changes in the lung tissue. The differentially expressed proteins identified in our study are involved in various signaling pathways that contribute to pulmonary fibrosis by influencing inflammation, ECM remodeling, angiogenesis and fibroblast activation. Remodeling processes are often driven by pro-fibrotic factors released during inflammation, which can promote the excessive production of extracellular matrix proteins and scarring. In most cases, it is resolved, but there is an emerging post-COVID-19 group that is at risk of developing chronic pulmonary fibrosis. This study focuses on severely affected COVID-19 patients from the early phase of the pandemic, a group in which additional risk factors, such as advanced age and smoking, may have heightened inflammatory responses and influenced tissue remodeling, mirroring the patterns observed in IPF. The study’s limitations include missing data on risk factors for some individuals and a small sample size, impacting statistical power. However, the unvaccinated status of this cohort offers essential insights into inflammatory and remodeling responses which are valuable for understanding potential outcomes in future pandemics. Guided by the hypothesis that chronic pulmonary diseases propagate through remodeling processes closely associated with inflammatory mechanisms, this analysis of unique early pandemic cases in Sweden reveals several markers overlapping with established IPF biomarkers. Markers found to be significantly elevated in COVID-19 compared to control samples included CCL19, CXCL-13, MCP-3, PGF, HGF and TNF. Of interest, DCN and TNFRSF12A were distinctly elevated in severe COVID-19, separating severe disease from moderate, while MCP-3 and HGF were found to increment with disease severity. Notably, 32 proteins were found to be significantly altered in the IPF cohort compared to healthy controls in our previous study but were not significantly altered in the COVID-19 study group. Among these were IL6 and angiopoietin-2 (ANGPT2), two proteins recognized as potential biomarkers for IPF.

The TNFRSF12A signaling pathway has been identified in IPF to accelerate collagen synthesis through the regulation of matrix metalloproteinase 9 (MMP9), which has been suggested as a therapeutic target in several lung diseases [29]. In a previous study, we discovered that TNFRSF12A was significantly elevated in IPF serum compared to healthy subjects and strongly correlated to disease progression [28]. Interestingly, in moderate COVID-19, the ligand to TNFRSF12A, TWEAK, was significantly reduced compared to in controls (Table 2), and although this was not significant, it was also reduced in the ICU group. This finding indicates that TNF signaling is increased in severe disease compared to moderate. While TNF signaling is crucial for tissue repair and the recruitment of cells such as fibroblasts, excessive or prolonged signaling is thought to contribute to fibrosis [30]. In a recent study by Iosef et al., patients with long COVID were found to have elevated plasma levels of TNF, TGF-β1 and MMP9 along with increased levels of angiogenic factors such as vascular endothelial growth factor A (VEGFA) and angiopoietin 1 (ANGPT1), indicating activated TNF-α pathway signaling and sustained involvement of the vascular system [31]. The inflammatory role of TNF in IPF has long been investigated [8]. In a study by Oikonomou et al., the link between TNF signaling and TGF-β1 expression was investigated, and the results demonstrated its involvement in fibrosis [30]. Thus, it is conceivable that the previously reported cytokine storm in severe COVID-19 predisposes for lung fibrosis [2,6], as we found the TNF-α pathway to be involved through TNFRSF12A.

Extracellular proteins linked to tissue remodeling, namely POSTN and DCN [20,32], along with chemokine CXCL13 [22], have all been proposed as prognostic biomarkers for IPF. Interestingly, all of these were altered in the COVID-19 patients compared to the healthy subjects. DCN was upregulated in the plasma from patients with severe COVID-19, differentiating them from patients with moderate COVID-19. In its soluble form, DCN has been proposed to prevent lung fibrosis in severe COVID-19 by engaging the receptor of interferon regulatory factors (IRFs), thereby protecting epithelial cells from apoptosis and also neutralizing TGF-β, a strong fibrosis mediator [14,32]. Meanwhile, ECM-incorporated DCN plays an important role in collagen fibrillogenesis. POSTN, on the other hand, a crucial factor for fibroblast activation, was found to be elevated in patients with both severe and moderate COVID-19 compared to controls, which has been shown by others as well to be associated with critical illness [33] and has been suggested as a potential therapeutic target in IPF [24]. Factors frequently discussed in IPF research including MMP7, MMP12, VEGFA and ANGPT2 did not show a significant change compared to control samples in our study.

We also found increased levels of CXCL13, a chemokine expressed by a subset of T helper cells upon antigen presentation and proposed as a biomarker for IPF [34]. We recently demonstrated in a previous study that the fibrotic environment of an IPF lung triggered a profibrotic response in healthy cells, where the chemokine CXCL13 was significantly elevated compared to cells cultured in a healthy lung microenvironment [10]. The combined increased levels of CXCL13 and HGF were shown in a large Swiss study to be the best predictor of COVID-19 admission to the intensive care unit (ICU) [35]. HGF, fundamentally involved in tissue regeneration, is likely to be upregulated to counteract the cyto/chemokine storm, including CXCL13, in an attempt to counteract fibrotic signals by interfering with TGF-β signaling. Further, the angiogenic factor PGF, correlated in one study to in-hospital mortality in COVID-19 [36], was, along with CCL19, GAL-9 and DCN, significantly elevated in patients with severe COVID-19 in comparison to healthy controls, as in our IPF ex vivo model and in IPF serum [28]. Interestingly, the two patients that died in hospital in our cohort had higher levels of CCL19 and PGF compared to the rest in the severe COVID-19 group. Furthermore, several monocyte attractant chemokines have been reported to be associated with disease severity in both COVID-19 and IPF patients. In our data, MCP-3 was an indicator of disease severity, which is in line with Yang et al.’s study [23] showing that there is a sustained high level of MCP-3 in the lung tissue in fatal cases even after the viral infection is gone. Taken together, these data emphasize the common pathways in IPF and severe COVID-19, which may explain the increased severity of COVID-19 infection in IPF patients [37]. In line with this, in patients with severe COVID-19, alveolar macrophages primarily interact with fibroblasts via the TNFSF12-TNFRSF12A signaling pathway, which drives fibroblast proliferation and the production of fibrotic factors [38]. TNFa, which is increased during severe COVID-19 infection, induces CXCL13, a predictor of IPF severity, in alveolar macrophages [39]. Based on our data, we therefore conclude that IPF patients are at risk of adverse outcome from COVID-19 infection.

## 5. Conclusions

Herein, we describe shared histopathological findings in explanted lung tissue from post-COVID-19 and IPF patients linked to ECM remodeling. Blood samples collected two weeks post-hospitalization from pre-vaccine COVID-19 patients exhibited elevated plasma levels of proteins associated with both tissue remodeling and inflammation that overlap with markers found in IPF, processes that are tightly connected. With time, hospitalization due to COVID-19 infection has decreased and only a few individuals have required transplantation in Sweden so far. However, there is an emerging subset of post-COVID-19 cases experiencing lung complications [40]. We found the combination of the following factors to be of most interest to follow up on in a larger cohort of post-COVID-19 patients: CXCL13, DCN, HGF, MCP-3 and TNFRSF12A. Notably, we lacked smoking status data for most of the severely affected COVID-19 patients, which may have influenced their inflammatory response. Although the direct link between COVID-19 and the development of IPF remains unestablished and is a subject of ongoing debate, these proteins may serve as critical biomarkers for identifying patients at risk of pulmonary fibrosis. This potential role underscores the importance of investigating their expression in larger, more diverse cohorts to validate their utility and relevance.

## Figures and Tables

**Figure 1 biomedicines-12-02893-f001:**
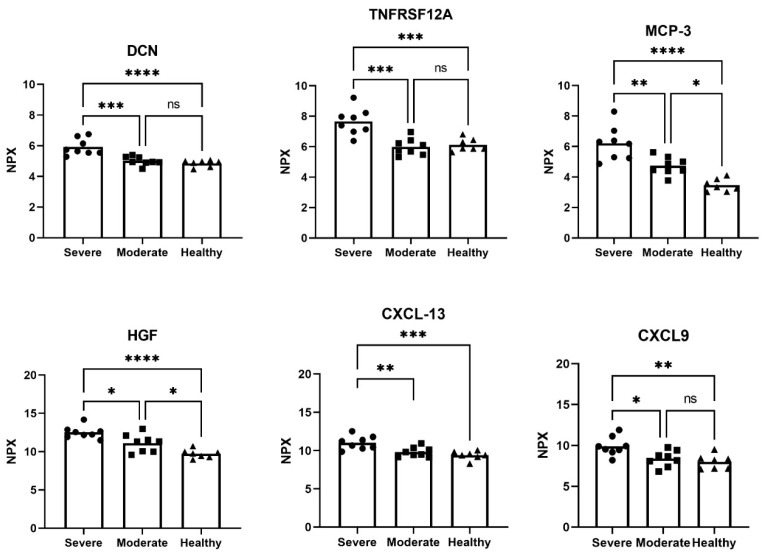
Elevated protein amount of DCN, TNFRSF12A, MCP-3, HGF, CXCL13 and CXCL9 in plasma from patients with moderate and severe COVID-19 in comparison to healthy subjects. NPX = normalized protein expression. Patients with moderate (n = 8) and severe (n = 8) COVID-19; healthy individuals (n = 7). One-way ANOVA with Tukey’s multiple comparison test. * *p* < 0.05, ** *p* < 0.01, *** *p* < 0.001, **** *p* < 0.0001, ns=not significant.

**Figure 2 biomedicines-12-02893-f002:**
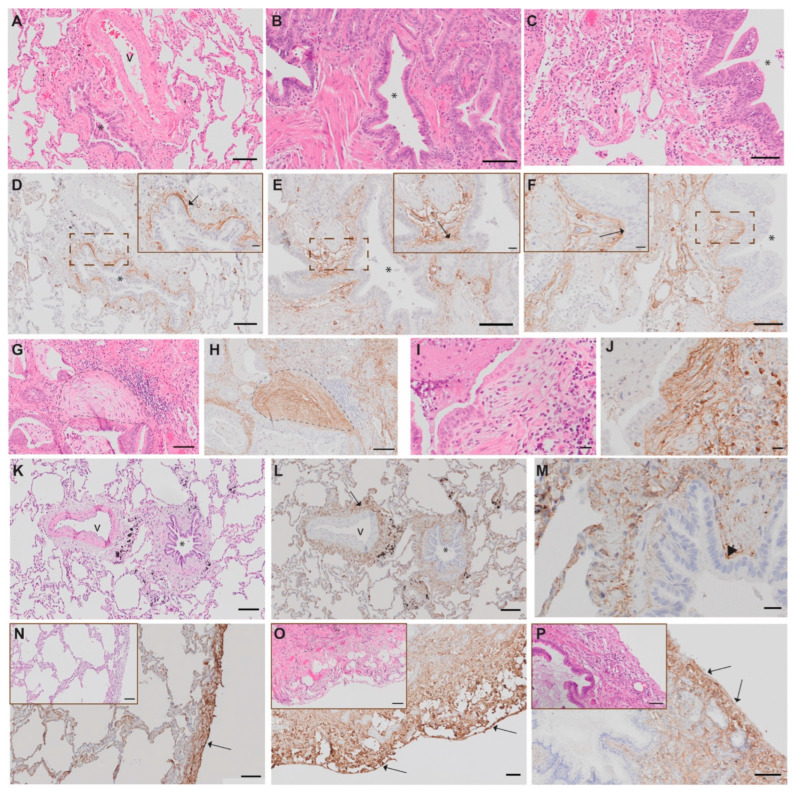
The overlapping protein patterns of DCN and POSTN in post-COVID-19 and IPF. In distal lung tissue, the expression of POSTN was mainly localized to the subepithelial regions of bronchioles in healthy (**A**,**D**), post-COVID-19 (**B**,**E**) and IPF (**C**,**F**) patients, enclosed upon magnification in the basement membrane zone (arrow). POSTN was highly expressed in fibroblastic foci in IPF (**G**,**H**, encircled area) and in similar structures in post-COVID-19 patients (**I**,**J**). Similarly, DCN was found to be intensely expressed in the subepithelial regions of bronchioles (arrowhead) and in vascular adventitia (arrow) (healthy, **K**–**M**). Increased DCN expression was also seen in pleura (arrows) and subpleural regions in healthy (**N**, including HE staining), post-COVID-19 (**O**, including HE staining) and IPF (**P**, including HE staining) patients. Scale bar: 500 µm (**N**–**P**); 100 µm (**A**–**F**, **K**–**M**; enlargements **N**–**P**); 20 µm (enlargement (**D**–**F**), **G**–**J**). * = bronchiole; v = vessel.

**Table 1 biomedicines-12-02893-t001:** Patient characteristics of plasma cohorts.

	Moderate COVID-19 (n = 8)	Severe COVID-19 (n = 8)	Healthy Subjects (n = 7)
Age (Mean ± SD)	57.1 ± 6.7	64.9 ± 13.8	55.1 ± 12.8
Male/female	5/3	7/1	5/2
(n, %)	(62.5%/37.5%)	(87.5%/12.5%)	(71%/29%)
Smoking history			
Never smoked (n, %)	6 (75%)	2 (25%)	3 (42.9%)
Ex-smokers (n, %)	2 (25%)	-	2 (28.6%)
Current smokers (n, %)	-	1 (12.5%)	2 (28.6%)
Unknown (n, %)		5 (62.5%)	
Death during hospital visit (n, %)	0 (0%)	2 (25%)	NA

**Table 2 biomedicines-12-02893-t002:** Altered plasma protein expression in patients with moderate and severe COVID-19 versus healthy subjects.

Protein	Mean Difference (NPX)	SE of Mean Difference	*p*-Value	Biological Process
Severe vs Healthy				
CCL19	1.512	0.5816	0.0433	I
LAG3	1.330	0.3436	0.0026	I
LAMP-3	1.259	0.3815	0.0095	I
KLRD1	1.185	0.3737	0.0127	I
IL15	1.179	0.2884	0.0016	I
CD70	1.045	0.2492	0.0012	I
GAL-9	1.099	0.2206	0.0002	I
ADGRG1	1.705	0.5260	0.0093	R
CAIX	1.404	0.4422	0.0127	R
PGF	1.284	0.3599	0.0052	R
DCN	1.065	0.1967	<0.0001	R
PDGFB	−1.683	0.5761	0.0220	R
TNF	1.110	0.3078	0.0280	O
PTN	2.107	0.5692	0.0039	O
MCP-1	1.451	0.3251	0.0007	O
Severe vs Moderate				
CXCL9	1.525	0.5082	0.0185	I
MCP-3	1.468	0.3895	0.0033	I
CXCL13	1.204	0.3540	0.0076	I
HGF	1.388	0.4436	0.0140	R
ADGRG1	1.167	0.5082	0.0719	R
PTN	1.108	0.5499	0.0030	R
TNFRSF12A	1.663	0.3250	0.0001	O
Moderate vs Healthy				
CD40	−1.882	0.3263	<0.0001	I
PD-L1	−1.752	0.3044	<0.0001	I
CD4	−1.077	0.3336	0.0112	I
TWEAK	−1.144	0.2957	0.0026	O

Significantly altered protein levels with ≥1 NPX mean difference (normalized protein expression, NPX), categorized according to biological function: I = inflammation/chemotaxis, R = tissue remodeling and O = overlapping functions. Standard error (SE) of difference in means.

## Data Availability

The data that support the findings of this study are available on request from the corresponding author. The data are not publicly available due to privacy or ethical restrictions.

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
