# Peer review of "Overlapping Systemic Proteins in COVID-19 and Lung Fibrosis Associated with Tissue Remodeling and Inflammation"

_biomedicines, 2024, doi:10.3390/biomedicines12122893_

Round 1
Reviewer 1 Report (Previous Reviewer 2)
Comments and Suggestions for Authors
Svobodová et al compared the 92 protein profile of COVID19 and IPF (Idiopathic Pulmonary fibrosis) patients and identified several proteins that are common to both the diseases. Authors expected to develop protein biomarkers that would predict whether a COVID-19 patients would develop IPF. This is interesting study and should be evaluated in larger cohorts.
However, following concerns should be taken care.
1. Authors stated they used a PEA panel of 92 proteins. Authors earlier published (ref 10) the proteins profile in IPF. Did they used the same panel? Is there any proteins that are altered in IPF vs healthy but not in COVID-19. Authors should discuss it.
2. Authors should discuss what would be the consequences of identified proteins if an IPF patients are infected with COVID-19.
3. Line 95, authors should expand P/F ratio
4. In supplementary figure S1, for CCL19 bargraph, from where “mild” patients are mentioned that are not described in methods.
Author Response
Dear Reviewer,
Thank you for thoroughly reviewing our manuscript and for your insightful comments. Below, we provide detailed responses to each of your points. We appreciate your feedback, which has helped us refine and strengthen our work.
---
Svobodová et al compared the 92 protein profile of COVID19 and IPF (Idiopathic Pulmonary fibrosis) patients and identified several proteins that are common to both the diseases. Authors expected to develop protein biomarkers that would predict whether a COVID-19 patients would develop IPF. This is interesting study and should be evaluated in larger cohorts.
However, following concerns should be taken care.
- Authors stated they used a PEA panel of 92 proteins. Authors earlier published (ref 10) the proteins profile in IPF. Did they used the same panel? Is there any proteins that are altered in IPF vs healthy but not in COVID-19. Authors should discuss it.
Response: Thank you for a good comment. Yes, it was the same panel of 92 proteins as in the earlier published study. Of the significantly altered proteins in the IPF cohort compared to healthy controls, there were 32 that were not altered in the COVID-19 study group. To mention a few that are known to be involved in pulmonary fibrosis:
- CXCL11: This chemokine is implicated in immune cell recruitment and has been associated with inflammatory responses in fibrotic lung diseases.
- IL6: A cytokine known to drive inflammation and fibrosis, including its role in pulmonary fibrosis.
- VEGFA (Vascular Endothelial Growth Factor A): Plays a role in angiogenesis and vascular remodeling, processes involved in fibrotic lung diseases.
- IL10: An anti-inflammatory cytokine; while not directly fibrotic, its regulatory role in inflammation is relevant in fibrosis contexts.
- ANGPT2 (Angiopoietin-2): Involved in vascular permeability and angiogenesis, contributing to fibrotic tissue remodeling.
- FGF2 (Fibroblast Growth Factor 2): Stimulates fibroblast activity and extracellular matrix production, playing a direct role in fibrosis development.
In the manuscript, we have added the following sentence on page 8, line 252-256: “Notably, 32 proteins were found to be significantly altered in the IPF cohort compared to healthy controls in our previous study (Elowsson Rendin et al. Int J Mol Sci 2019), but were not significantly altered in the COVID-19 study group. Among these were IL6 and angiopoietin-2 (ANGPT2), two proteins recognized as potential biomarkers for IPF.”
- Authors should discuss what would be the consequences of identified proteins if an IPF patients are infected with COVID-19.
Response: We appreciate the comment and have added in the manuscript on page 9, lines 308-315.
Taken together, these data emphasize the common pathways in IPF and severe COVID-19, which may explain the increased severity of COVID-19 infection in IPF patients (37). In line, in severe COVID-19 patients, alveolar macrophages primarily interact with fibroblasts via the TNFSF12-TNFRSF12A signaling pathway, which drives fibroblast proliferation and the production of fibrotic factors (38). TNFa, which is increased during severe COVID-19 infection, induces CXCL13, a predictor of IPF severity, in al-veolar macrophages(39). Based on our data, we therefore conclude that IPF patients are might be of risk of adverse outcome from COVID-19 infection.
- Line 95, authors should expand P/F ratio
Response: Thank you for noticing. We have changed to PaOâ‚‚/FiOâ‚‚ ratio.
- In supplementary figure S1, for CCL19 bar graph, from where “mild” patients are mentioned that are not described in methods.
Response: Thank you for noticing. It was a mistake and has been corrected with ‘moderate’.
Reviewer 2 Report (Previous Reviewer 1)
Comments and Suggestions for Authors
I sincerely appreciate the author’s perspective as outlined in their response to the reviewers. However, I have several concerns that remain unresolved:
-
The refusal to present the ROC curve raises significant concerns about the transparency and robustness of the data. In accordance with open data practices, it is imperative to provide individual data values to ensure reliability and reproducibility.
-
The reluctance to perform double-staining IHC further undermines confidence in the data's validity. Tissue autofluorescence is a well-known issue that has been effectively addressed through standard methodologies. The use of antibodies with XTITC/TRITC and Cy5.5 excitation channels is a widely accepted approach to eliminate autofluorescence and would have strengthened the findings significantly.
Given that these critical points were not adequately addressed, I regret that I cannot recommend the paper for publication in its current form.
Author Response
Dear Reviewer,
Thank you for thoroughly reviewing our manuscript and for your insightful comments. Below, we provide detailed responses to each of your points. We appreciate your feedback, which has helped us refine and strengthen our work.
---
I sincerely appreciate the author’s perspective as outlined in their response to the reviewers. However, I have several concerns that remain unresolved:
- The refusal to present the ROC curve raises significant concerns about the transparency and robustness of the data. In accordance with open data practices, it is imperative to provide individual data values to ensure reliability and reproducibility.
Response: We understand the reviewer’s concerns regarding the absence of ROC curves. We have investigated this further, and have been in contact with a biostatistician expert at Olink, where we performed the protein analysis. They emphasized that the limited size in this study reduces the statistical robustness of such an analysis, as small cohorts are prone to variability and potential overfitting. Including ROC curves under these conditions could lead to misinterpretation or overestimation of the predictive value of the markers analyzed. Our study is constrained by investigating a small group of patients from the Swedish population before the initiation of vaccination programs, further limiting generalizability. Therefore, we highlight the importance of investigating our findings in larger cohorts to robustly validate the predictive potential of these proteins as biomarkers. See page 10, lines 328-332
“Although the direct link between COVID-19 and the development of IPF remains unestablished and is a subject of ongoing debate, these proteins may serve as critical biomarkers for identifying patients at risk of pulmonary fibrosis. This potential role underscores the importance of investigating their expression in larger, more diverse cohorts to validate their utility and relevance.”
- The reluctance to perform double-staining IHC further undermines confidence in the data's validity. Tissue autofluorescence is a well-known issue that has been effectively addressed through standard methodologies. The use of antibodies with XTITC/TRITC and Cy5.5 excitation channels is a widely accepted approach to eliminate autofluorescence and would have strengthened the findings significantly.
Response: We appreciate the reviewer’s suggestion to adopt double-channel immunofluorescence for co-localization analysis. However, in this study, DAB-based IHC was chosen for its superior ability to delineate tissue morphology and pathological features, which were central to our investigation. Our aim was to broadly map the distribution of proteins, which DAB staining effectively achieves. The insights gained from DAB IHC align with our hypothesis regarding their roles in lung fibrosis, even without detailed co-localization. Nonetheless, we recognize the utility of immunofluorescence and is something we are considering for future studies.
Reviewer 3 Report (New Reviewer)
Comments and Suggestions for Authors
The authors identified systemic proteins that overlap with fibrotic markers identified in patients with idiopathic pulmonary fibrosis that may predict COVID-19-induced lung fibrosis. I think this manuscript is well written and worthwhile to be published in this journal.
Author Response
The authors identified systemic proteins that overlap with fibrotic markers identified in patients with idiopathic pulmonary fibrosis that may predict COVID-19-induced lung fibrosis. I think this manuscript is well written and worthwhile to be published in this journal.
Response: We thank the reviewer for the recognition of the value of our manuscript and their kind comments.
Round 2
Reviewer 2 Report (Previous Reviewer 1)
Comments and Suggestions for Authors
The authors have disregarded significant questions raised about their open-data practices and have refused to address them, raising serious ethical concerns.
Additionally, the authors failed to respond to inquiries regarding double staining, which undermines the validity of their findings and the methods employed.
This manuscript is a resubmission of an earlier submission. The following is a list of the peer review reports and author responses from that submission.
Round 1
Reviewer 1 Report
Comments and Suggestions for Authors
The authors describe changes in plasma protein samples of COVID patients, related to the pre-vaccination period, with the aim of identifying specific prognostic markers. Some data are complemented with lung tissue histology analysis.
While the methodology of the study is sound and complex, and the findings are valuable, they do not significantly advance our understanding of the topic, as most of the detected proteins are related to extracellular matrix remodeling and/or inflammation as it was expected for COVID and was shown in many other reports.
The conclusions of the study are also not fully justified when decorin (DCN) and TNFRSF1A are used for "separation" of disease forms, as there are no ROC curves or AUROC values presented.
The identified protein values have overlapping ranges between groups, further questioning their usefulness in predictive approaches.
Moreover, DCN and periostin (POSTN) are usually considered to have opposite roles in fibrosis development, and thus the immunohistochemistry (IHC) study with DAB staining and referring to "patterns of staining" is not relevant. A double-channel immunofluorescence approach is needed to determine whether areas of increased expression overlap in the same lung tissue or if there are distinct pro- and anti-fibrotic clusters within the same organ.
Providing full names for ECM, DCN in the abstract and a brief description of these, as well as periostin in the text, will make the manuscript more engaging for the reader.
Unfortunately, due to the lack of novelty and appropriate justification of claims I can not recommend the current manuscript to be published.
Reviewer 2 Report
Comments and Suggestions for Authors
Svobodová et al performed the protein profiling to distinguish COVID-19 induced pulmonary fibrosis from IPF . They used relatively a new technique PEA for measuring 92 proteins in the serum samples from moderate and severe COVID-19 patients. By immunohistology, they showed alteration of localization of DCN and POSTN in explanted lung tissue. They identified five proteins (DCN, TNFRSF12A, CXCL13, MCP-3 and HGF) plays vital role in advancement of pulmonary fibrosis in COVID-19 patients and could be studied further.
These are the following concerns that should be modified/corrected.
1. In the abstract, ECM should be expanded.
2. Line 99, heading should be “explanted lung tissue or something like that rather than “human material” because plasma is also human material.
4. PEA is a new but a powerful technique for protein estimation, they should write the principle and procedure in the methods.
5. Authors should mention on what basis they selected 92 proteins for the assay.
6. Line 188, Authors stated that they had shown the role of POSTN in IPF lung tissue, but did not show the result in PEA assay of COVID-19 sample. Is it not significant? If it is not significant, they should mention it with an explanation. Because there are much of the discussion about POSTN only based on immunohistochemistry data.
7. Line 234, what happened to TWEAK in severe condition? Is it increased or decreased. Without this information, the discussion of line 2333-235 does not explain the conclusion.
8. A network of differentially expressed proteins leading to the development of pulmonary fibrosis in COVID-19 would have been helpful.
9. 9. Did the authors observed any proteins that are highly involved in IPF but not altered in COVID-19.